# Cerebral Vein Thrombosis and Direct Oral Anticoagulants: A Review

**DOI:** 10.3390/jcm13164730

**Published:** 2024-08-12

**Authors:** Johanna Umurungi, Francesca Ferrando, Daniela Cilloni, Piera Sivera

**Affiliations:** 1Department of Clinical and Biological Sciences, University of Turin, 10043 Turin, Italy; daniela.cilloni@unito.it; 2Department of Medical Sciences, University of Turin, 10126 Turin, Italy; francesca.ferrando@unito.it; 3Haematology and Cellular Therapies Unit, A.O. Ordine Mauriziano di Torino, 10128 Turin, Italy; psivera@mauriziano.it

**Keywords:** intracranial thrombosis, anticoagulants, antithrombins, factor Xa inhibitors

## Abstract

Cerebral venous thrombosis (CVT) is a rare type of cerebrovascular event in which the thrombosis occurs in a vein of the cerebral venous system. The diagnosis could be challenging due to the great clinical variability, but the outcome is favourable in most cases, especially in the case of early diagnosis. Anticoagulant therapy is the core of CVT management and currently consists of heparin in the acute phase followed by vitamin K antagonists (VKAs) in the long term. The ideal duration of anticoagulant therapy is still unclear, and the same criteria for the treatment of extracerebral venous thromboembolism currently apply. In this paper, we reviewed the literature regarding the use of direct oral anticoagulants (DOACs) in CVT since in recent years a considerable number of studies have been published on the use of these drugs in this specific setting. DOACs have already been shown to be equally effective with VKAs in the treatment of venous thromboembolism. In addition to efficacy, DOACs appear to have the same safety profile, being, on the other hand, more manageable, as they do not require close monitoring with continuous personalised dose adjustments. In addition, a further advantage of DOACs over VKAs is the possibility of anticoagulant prophylaxis using a reduced dosage of the drug. In conclusion, although the use of DOACs appears from preliminary studies to be effective and safe in the treatment of CVT, additional studies are needed to include these drugs in the treatment of CVT.

## 1. Background

### 1.1. Epidemiology

Cerebral venous thrombosis (CVT) is a cerebrovascular disease caused by the partial or complete occlusion of one or more veins constituting the cerebral venous system. Unlike arterial stroke, CVT affects women, children and young adults more frequently. Its incidence is estimated to be 1–2/100,000/year in Western Europe, and mortality is below 5% [1].

### 1.2. Pathogenesis

CVT can affect the superficial, deep cerebral veins or the dura mater sinuses. Isolated thrombosis of the superficial veins is rare, and the involvement of the superficial veins is usually caused by thrombosis of the dural venous sinuses. Thrombosis of the deep cerebral veins usually involves the internal cerebral vein and the vein of Galen [2,3]. Frequently, CVT affects multiple dural venous sinuses, most commonly affecting the transverse sinus and the superior sagittal sinus (Figure 1) [4]. The venous sinuses of the dura mater constitute a complex network of venous channels that drain blood from the brain and cranial bones. They develop from the venous plexuses and are included in the dura mater splits. Unlike the other veins, they have no valves (thus, the direction of flow is reversible) and lack smooth muscle tissue in the constitution of the walls. Considering the anatomical variability and multiple anastomoses between the cerebral veins and dural sinuses, clinical manifestations are very heterogeneous, varying from asymptomatic condition to death depending on the site of the thrombosis and the collateral circles [1,3,5,6].

Two different mechanisms seem to contribute to the clinical manifestation [3,6]:Thrombosis of the cerebral veins can cause localised oedema of the brain and venous infarction. Two different types of cerebral oedema can develop. The first is a vasogenic oedema caused by an increase in veno-capillary pressure and the rupture of the blood–brain barrier with consequent extravasation of blood into the interstitium. Vasogenic oedema is reversible if the underlying condition is treated. The second is a cytotoxic oedema caused by ischaemia that damages cellular membrane pumps, resulting in intracellular swelling.Occlusion of the dural sinuses with reduced cerebrospinal fluid reabsorption and increased intracranial pressure (more frequent in the presence of sagittal sinus thrombosis).

In approximately 85% of cases of cerebral venous thrombosis, a direct cause or a prothrombotic risk factor can be identified. Often, a precipitating factor (e.g., head trauma) can trigger the event in patients who already have a thrombotic predisposition. Predisposing factors are multiple and are classically related to Virchow’s triad (blood stasis, vessel wall alterations and blood hypercoagulability), but CVT is not associated with classic arterial risk factors. Risk factors and predisposing conditions are usually classified into acquired and congenital, and they can be transient or chronic conditions (Table 1) [2,3,4,6]. The most frequently acquired prothrombotic conditions are gender-specific: pregnancy, puerperium and the use of oestrogen-containing contraceptives. Other conditions include obesity (particularly when associated with the use of oral contraceptives), antiphospholipid antibody syndrome and, more rarely, infections (head-neck infections, COVID-19, sepsis), head trauma, inflammatory diseases (systemic lupus erythematosus, Behçet’s disease, chronic inflammatory bowel disease), haematological disorders (mutated JAK V617F myeloproliferative diseases, paroxysmal nocturnal haemoglobinuria and haemoglobinopathies) and pro-thrombotic medications (corticosteroids, L-aspariginase, thalidomide-analogues, tamoxifene). Finally, in patients over 55 years of age, the prevalence of CVT is the same in both sexes, and in 25% of cases, the cause is an underlying malignancy.

The congenital risk factors associated with the development of CVT are hereditary thrombophilia, including the factor V Leiden mutation, the prothrombin polymorphism (G20210A mutation) and congenital antithrombin, coagulative protein C and protein S deficiency [2,3,4]. However, in a minority of cases the origin remains unknown, and this is particularly the case in elderly individuals over 65 years of age [7].

### 1.3. Clinical Manifestations

The clinical presentation of CVT is highly variable due to the multitude of possible underlying conditions, and the onset may be acute, subacute or chronic. In general, the signs and symptoms of CVT can be grouped into three main clinical presentations: isolated intracranial hypertension syndrome (with headache, possibly vomiting, papilledema and visual disturbances); focal syndrome (motor or sensory impairments, aphasia, hemianopsia, focal seizures); encephalopathy (multifocal signs, altered state of consciousness, stupor or coma) [5,8,9]. Patients usually show symptoms and signs related to both intracranial hypertension and focal syndromes at presentation or with the progression of the underlying disease. Headache, usually a sign of cerebral hypertension, is the most common symptom and occurs in more than 90% of cases. The headache in this case is typically diffuse and progressive over time, but in a minority of patients, it may occur as a thunderclap headache or as a migraine. Focal neurological signs and symptoms are often present in cases of ischaemia or venous haemorrhage with focal brain damage and are attributable to the affected area. In addition, the symptomatology may also be related to the underlying cause. For example, in the case of lateral sinus thrombosis, an ipsilateral middle ear infection can often be found, with signs and symptoms related to the infection itself. Finally, certain features help distinguish CVT from other cerebrovascular conditions, including the frequent presence of focal or generalised seizures, bilateral brain involvement and slow progression of symptoms [2].

### 1.4. Diagnosis

Due to the great clinical variability, the diagnosis of CVT can be very difficult and is normally based on a clinical suspicion that is then confirmed by neuroimaging tests. In patients presenting with new-onset neurological symptoms, the first assessment is often performed by Computed Tomography (CT). Due to the anatomical variability of the cerebral venous circulation, CT has a low sensitivity in the study of CVT, and a CT without contrast is altered in only about 30% of CVT cases. The first sign of CVT in non-contrast CT is a hyperdense signal of a cortical vein or dural sinus, but it is found only in one-third of CVT. More rarely, signs of subarachnoid haemorrhage or intracranial haemorrhage can be found. Contrast-enhanced CT may show instead a filling defect within the vein or sinus [1,2]. Overall, the accuracy in the diagnosis of CVT by CT combined with contrast-enhanced CT ranges from 90 to 100% depending on the site of occlusion [10].

Another neuroimaging technique that can be used is magnetic resonance imaging (MRI), which is, in general, more sensitive than CT. MRI is, in fact, able to visualise the thrombus within a venous sinus and the characteristics of the images will depend on the age of the thrombus itself [11]. It is also able to identify indirect signs of CVT such as oedema, haemorrhage or parenchymal lesions better than CT. Finally, enhanced-contrast MRI is even more sensitive in identifying the absence of flow in a cerebral venous sinus even though that image interpretation can be difficult due to the presence of normal anatomical variants such as sinus hypoplasia or the presence of asymmetric flow [2]. Despite the greater sensitivity of MRI, it must be remembered that this technique is not always available, and therefore, in these cases, CT venography is considered a valid alternative for the diagnosis of CVT [1,10]. 

Finally, in cases where the diagnostic suspicion for CVT is high but there has been no CT or MRI finding, there are available invasive angiographic diagnostic procedures that allow the direct visualisation of flow interruption within a vessel [2].

In addition to the imaging evaluation, then, a laboratory evaluation must always be associated with the performance of tests not to confirm the diagnosis but to search for possible triggering factors or underlying causes. Laboratory tests should include a complete blood count, biochemistry panel, indices of inflammation and coagulation panel (prothrombin time, activated partial thromboplastin time, D-Dimer, fibrinogen). These examinations could reveal the presence of underlying prothrombotic, inflammatory or infectious processes. Regarding D-Dimer, a few small studies have demonstrated a high sensitivity for identifying patients with CVT and its potential negative predictive value (as for deep vein thrombosis and pulmonary embolism), but this finding is not universally accepted, and a normal D-Dimer value cannot exclude CVT. Screening for congenital and acquired thrombophilia should only be carried out in patients with a high probability of being affected by thrombophilia, i.e., young patients and those with a positive personal or family history of thromboembolic events, especially if young and in cases of apparently idiopathic CVT. In contrast, in patients over 40 years of age without evidence of clear aetiology, an occult neoplasm and/or a haematological prothrombotic disease (such as myeloproliferative syndromes) should always be searched. Finally, lumbar puncture for the evaluation of cerebrospinal fluid is not usually helpful unless there is a concomitant suspicion of meningitis [1,2,3].

### 1.5. Prognosis

CVT tends to have a good outcome with a favourable prognosis in about 75% of cases. Nevertheless, 15% of patients die or have residual disability. Acute mortality has been reduced in recent years thanks to improved therapies and diagnostic techniques that allow for greater identification of even the least severe cases and is currently around 5% [5,12,13]. Predictors of acute mortality include impaired consciousness, deep vein thrombosis, right hemisphere haemorrhage and posterior fossa injuries. The main causes of acute mortality are transtentorial herniation due to massive haemorrhage, diffuse cerebral oedema, generalised sickness and medical complications [5,14]. With regard to long-term outcomes, predictive factors include central nervous system infections, malignancy, deep vein thrombosis, cerebral haemorrhage, altered state of consciousness, age over 37 and male gender [5].

### 1.6. Therapy

Once the diagnosis of CVT has been confirmed, therapy should be started as soon as possible. Therapy includes the initiation of anticoagulants, treatment of any underlying cause, seizure control and management of endocranial hypertension, if necessary [1,15].

Anticoagulant therapy aims to avoid thrombus extension and promote thrombus resolution but at the same time could cause or worsen intracranial haemorrhages. From the studies carried out, however, anticoagulants appear to be safe in adult patients with CVT who have associated intracranial haemorrhage [15,16].

In the acute phase of CVT, the anticoagulant therapy of choice is heparin, preferably low molecular weight (LMWH). The use of unfractionated heparin (UFH) should be considered in the case of unstable patients who are likely to undergo surgery or are awaiting lumbar puncture [3,15].

During the acute phase, a thrombolysis strategy may also be considered, which, unlike anticoagulant therapy, aims to immediately reduce the size of the thrombus by topically administering fibrinolytic agents or mechanically removing it [17,18]. There appears to be a good rate of recanalisation, but it is associated with a non-negligible rate of intracranial haemorrhage without a statistically significant difference in outcomes compared with anticoagulant therapy [3]. For this reason, endovascular treatment is only considered in cases of severe CVT that do not improve or worsen despite anticoagulant therapy and in patients with a high pre-treatment risk of poor outcome [1,15].

Long-term anticoagulant therapy is indicated to prevent further thrombotic events: the risk of CVT recurrence is about 2–7% per year, and the risk of other venous thrombosis is 4–7% per year [3,5]. For long-term therapy, guidelines recommend using vitamin K antagonists (VKAs) maintaining the standard range of INR (international normalised ratio) between 2.0 and 3.0. The optimal duration of anticoagulant therapy after the acute phase is not yet clearly defined. At present, the same criteria generally apply as for extracerebral thrombosis: if CVT is due to a transient risk factor, anticoagulant therapy is recommended for 3 months; if CVT is apparently idiopathic and/or if it is associated with thrombophilia, therapy should be continued for 6 to 12 months; sine die anticoagulant therapy should be considered in the case of patients with recurrent CVT or associated with thrombophilia at high thrombotic risk [1,15].

## 2. Direct Oral Anticoagulants

The direct oral anticoagulants (DOACs) are a relatively new category of anticoagulant drugs recognised as an effective and safe alternative to VKAs in the treatment of atrial fibrillation and venous thromboembolism (deep vein thrombosis of the leg and pulmonary embolism). Moreover, these drugs are more manageable, as they do not require INR monitoring or dose adjustment, do not require dietary restrictions and have a lower rate of intracranial haemorrhage than VKAs [14,19,20]. 

Despite this, current guidelines only consider VKAs an anticoagulant therapy for CVT, probably also because they predate the scientific evidence concerning the possible use of DOACs [1,16]. In recent years, particularly since 2017, a considerable number of studies have been published on the use of DOACs in the long-term treatment of cerebral venous thrombosis [21,22,23,24,25,26,27,28,29,30,31,32,33,34,35,36,37,38,39,40]. Although there is still not much evidence, both Warfarin and DOACs seem to be equally safe and effective in preventing recurrences of CVT and/or other forms of venous thromboembolism and, even, some studies suggest the achievement of a better level of anticoagulation with the use of DOACs vs Warfarin due to the difficulty in achieving and maintaining a therapeutic INR while using the latter [25,29,31].

In particular, ACTION-CVT, a retrospective observational study that included 845 patients with mild to moderate cerebral venous thrombosis (CVT) treated with Warfarin (52%) versus DOACs (33%) versus both at different times (15%) demonstrated substantial equivalence in recurrence rates and the percentage of ongoing bleeding during therapy with Warfarin versus Direct Oral Anticoagulants (DOACs). This study showed a similar risk of recurrence and death, as well as a similar rate of re-canalisation, but a lower risk of major bleeding with the use of DOACs [26].

Similar findings have emerged from various meta-analyses, systematic reviews and observational studies, confirming similar rates of mortality and morbidity with comparable or greater effectiveness in re-canalisation and recurrence prevention, with a lower or equal risk of major bleeding. These results do not appear to be affected by patient selection bias, as clinical factors did not drive the choice of one anticoagulant regimen over the other, resulting in homogeneity among groups in terms of disease severity. Finally, these findings appear to be applicable and true even in the subgroup of oncology patients and elderly patients (age > 80 years), with the limitations associated with a very small sample size.

### 2.1. Dabigatran

Dabigatran is a direct thrombin inhibitor, administered orally and with predominantly renal elimination (80%). The standard anticoagulant dosage is 150 mg bis in die (bid) and indications to reduce to a dose of 110 mg bid are increased haemorrhagic risk, age > 80 years, CrCl 30–50 mL/min and concomitant use with strong p-glycoprotein inhibitors (P-gp inhibitors) such as ketoconazole. In addition, in the USA, there is a further indication of dose reduction to 75 mg bid in the case of CrCl 15–29 mL/min. In contrast, its use is contraindicated in cases of CrCl < 15 mL/min, in severe liver failure and in concomitant use of P-gp inducers (carbamazepine, phenytoin and St. John’s Wort) [41,42,43]. Finally, it is important to remember that there is a specific antidote for this drug, idarucizumab, which is indicated if rapid reversal of the anticoagulant effect is required, for example, in the case of urgent surgery or in the event of uncontrolled and potentially fatal bleeding.

Regarding the use of dabigatran in the treatment of CVT, the literature on this topic includes some small-scale descriptive studies and a single, more structured RCT (Randomised Controlled Trial) by Ferro et al. In this trial, a total of 115 patients were treated with Dabigatran for 24 weeks after 5–15 days of therapy with LMWH/UFH. In the Dabigatran-treated group, there were no deaths or new intracranial haemorrhages (two in the Warfarin-treated group); only 7 patients discontinued the treatment (two due to episodes of major bleeding: abdominal hematoma, increased size of known intracerebral bleeding); similar values in terms of radiographic improvement of CVT (Cerebral Venous Thrombosis) and degree of disability according to the Modified Rankin Scale [32].

### 2.2. Apixaban

Apixaban is a direct inhibitor of factor Xa and, among DOACs, is the one with the lowest renal excretion and highest hepatic metabolism via cytochrome P450. The standard anticoagulant dose is 10 mg bid for the first 7 days and 5 mg bid thereafter, and a dose reduction is indicated if two of the following conditions are present: age ≥ 80 years, body weight ≤ 60 kg, serum creatinine ≥ 1.5 mg/dL. In Europe, however, the recommendations are for a standard dose if creatinine’s clearence (CrCl) > 30 mL/min, dose adjustment if CrCl is between 15 and 30 mL/min, and avoidance if CrCl < 15 mL/min. Apixaban is one of the most studied drugs in special populations and its use appears to be effective and safe even in cancer patients, in patients with mild to moderate hepatopathy and in cases of severe renal damage and the need for haemodialysis. In the case of uncontrolled or potentially fatal bleeding, the reversal therapy can be conducted with prothrombin complex concentrate or the specific antidote, andexanet alpha [41,42,43].

For the case of CVTs, the literature currently includes only case series [33,34] and a retrospective single-centre study of 9 patients [35]. The latter demonstrated good efficacy (78% complete recanalisation, 22% partial recanalisation) with no adverse bleeding events [35]. Similar results with improved recanalisation rates compared to Warfarin were found in a retrospective single-centre study by Christodoulides et al., in which the case group consisted of 26 patients treated with Apixaban or Rivaroxaban [31]. These data are, of course, limited by their retrospective, non-randomised nature, and small sample size; nevertheless, they suggest a favourable efficacy profile for the use of these agents.

### 2.3. Rivaroxaban

Rivaroxaban is a direct inhibitor of factor Xa, 65% of which is metabolised by the liver via cytochrome P450, and the remaining 35% is eliminated renally, which is why particular care must be taken in the case of concomitant use of CYP3A4 inhibitors. The anticoagulant dose is 15 mg bid for the first 21 days followed by 20 mg die. A dose reduction to 15 mg die is indicated in the case of CrCl of 15–49 mL/min and its use is contraindicated in Europe if CrCl < 15 mL/min. In the case of uncontrolled or potentially fatal bleeding, the anticoagulant effect may be reversed by administering prothrombin complex concentrate or the specific inhibitor andexanet alpha [41,42,43].

Despite clinicians showing a preference for the use of this molecule in the literature [36,37,38,39,40], to date, only a few small-scale retrospective studies and three prospective observational studies [36,37,40] can be found. Of these, one involves 31 patients and demonstrates a favourable safety and efficacy profile of Rivaroxaban as a chronic treatment following LMWH/UFH [40]. Another, on the other hand, evaluated the use of Rivaroxaban in the acute phase without prior use of LMWH/UFH in 20 clinically stable patients, also showing a good safety and efficacy profile in this setting (complete recanalisation 60%, partial 40%) [37]. The results of two studies on the use of Rivaroxaban versus standard-of-care for the treatment of symptomatic cerebral venous thrombosis are highly anticipated, as they are randomised trials that could give important additional information on the use of this drug in CVT (Study of Rivaroxaban for CeREbral Venous Thrombosis—SECRET, NCT03178864 at the University of British Columbia; Rivaroxaban vs Warfarin in CVT Treatment—RWCVT, NCT04569279 at Damascus University).

### 2.4. Edoxaban

Edoxaban is a direct inhibitor of factor Xa with renal and hepatic metabolism. The anticoagulant dose is 60 mg die, and a dose adjustment to 30 mg die is recommended if body weight < 60 kg, CrCl 15–49 mL/min or there is concomitant use of a potent P-gp inhibitor (cyclosporine, dronedarone, erythromycin or ketoconazole). Contrary to other DOACs, edoxaban has been shown to be less effective than warfarin in patients with CrCl > 95 mL/min, and for this reason, its use is not recommended in this condition [41,42,43]. 

To date, there are no studies of sufficient size and/or well-structured studies regarding the use of this type of DOAC (Direct Oral Anticoagulant) in the context of CVT (Cerebral Venous Thrombosis).

## 3. Our Experience

In our centre, twelve cases of cerebral venous thrombosis have been treated and observed from December 2017 to July 2024. It is important to specify that in Italy, and consequently in our centre, the use of DOACs as an acute treatment of CVT is not yet authorised, and since we have not conducted a clinical trial, we have not been able to use these drugs in the acute phase of the disease. Specifically, the 12 cases consisted of nine women and three men with a mean age of 41.1 years and 54.7 years, respectively. Regarding the aetiology, four cases were related to the use of oestrogen contraceptives, and one of these cases was also found to be weakly positive for the JAK2 mutation without a diagnosis of myeloproliferative disease; in two cases, there was a previous diagnosis of JAK2-positive myeloproliferative disorder (essential thrombocythaemia and polycythaemia vera); in two cases, there was an underlying congenital thrombophilia and an autoimmune disease (factor V Leiden mutation associated with Crohn’s disease and antithrombin deficiency associated with lichen ruben planus); one case arose following a head injury and, lastly, three cases were defined as idiopathic (Table 2).

Most of the patients, with the exception of three, had comorbidities or additional thrombotic risk factors. Some patients had more comorbidities and confounding risk factors, as in the case of patient ID1, who had a history of Budd–Chiari syndrome with a need for liver transplantation and a JAK2-positive essential thrombocythaemia not under active cytoreductive treatment. Other frequent comorbidities in the study population included obesity (3 out of 12 patients) and high blood pressure (3 out of 12 patients); two patients had autoimmune diseases: thyroiditis and Crohn’s disease. Of the patients’ pre-existing chronic therapies, except for oestrogen drugs considered to be triggers for CVT, the most commonly used drugs were antihypertensive drugs. Among others, one patient was taking mesalazine for Crohn’s disease, one patient was on ruxolitinib and oncocarbide therapy for polycythaemia vera and two patients were already on anticoagulant therapy with Coumadin.

With regard to clinical manifestations, most patients complained of intense and persistent headaches that were unresponsive to the usual antalgic therapies, and only a minority of them had focal neurological signs or symptoms. There were hardly any changes upon objective examination. The diagnosis was made in all cases by neuroimaging, mainly by magnetic resonance angiography (MRA). Finally, almost all patients had multiple venous sinus involvement (Table 3).

Regarding therapy, all our patients were treated with LMWH imbricated with warfarin with maintenance of an INR between 2 and 3, except one patient, who continued LMWH therapy by voluntary decision due to difficulties in the management of warfarin therapy. Thereafter, of the eleven patients who completed the one-year follow-up and received at least six months of warfarin therapy, four were placed on warfarin and one on acenocoumarol sine die therapy (one patient diagnosed with JAK2-positive essential thrombocythemia, one patient with AT-III deficiency, one patient with idiopathic thrombosis and progression after COVID infection and one patient had a previous diagnosis of polycythemia vera JAK2+ with no history of venous thrombosis), five started prophylaxis with DOACs (one patient due to the persistence of neurological symptoms and the finding of JAK2 positivity at low titre, in one case for congenital asymmetry of venous sinus calibre and in the three cases considered idiopathic) and two were placed on LMWH prophylaxis only in conditions of high thrombotic risk.

Ten cases completed follow-up at one year, counting six complete recanalisations, two partial recanalisation and two non-recanalisations, of which one included a progression of the thrombosis. Of the other two cases, one was lost to follow-up and the other has not yet been re-evaluated at one year but shows a non-recanalisation at six months (Figure 2).

## 4. Conclusions

DOACs have already been shown to be equally effective with VKAs in the treatment of venous thromboembolism. The most notable finding is a significant reduction in major bleeding during DOAC therapy, and this positive effect is maintained even in elderly patients and in patients with moderate renal insufficiency [20]. In addition, it must be considered that therapy with DOACs has important practical implications, as there is no need for close monitoring with continuous personalised dose adjustments [20,23]. Finally, a further advantage of DOACs over VKAs is the possibility of anticoagulant prophylaxis using a reduced dosage of the drug, and this would allow us, for example, to continue prophylaxis in patients with idiopathic CVT in whom it does not appear prudent to suspend anticoagulant therapy given the atypical site of the thrombosis or in those patients in whom there is an underlying thrombophilic condition that cannot be resolved.

Although several data suggest the superiority of DOACs over VKAs in the treatment of venous thromboembolism, their use is currently not yet recommended in guidelines for the management of cerebral venous thrombosis mainly due to a lack of evidence-based data [1,3,15]. This problem can only be overcome through randomised clinical trials involving an adequate number of subjects and comparing the use of DOACs with therapy based on LWMH and VKA, which are currently considered the standard of care. Unfortunately, there are no randomised studies investigating the use of DOACs in this setting. Furthermore, with the studies currently available it is not possible to conduct meta-analyses. 

In addition, there are two further aspects that need to be considered. The first is the possible interaction between DOACs and anticonvulsant drugs that are often used in the setting of CVT. 

It is now known, in fact, that most anti-epileptic drugs interfere with the metabolism of DOACs via cytochrome P450 and P-glycoprotein reducing their concentration in the blood and thus increasing the thrombotic risk [44,45].

The second, instead, concerns the possibility of being faced with a cancer patient for whom it is not possible to start anticoagulant therapy with warfarin, and, therefore, the standard of care in these patients is represented by LWMH therapy. Since 2018, however, the results of six randomised controlled trials have been published, showing that DOACs, compared to LWMH, significantly reduce the risk of thrombosis recurrence, with a non-significant increased rate of major bleeding but a significant increase of clinically relevant non-major bleeding [46]. Therefore, in these patients, the use of DOACs would be further legitimised while considering the subgroups of patients at increased risk of bleeding and therefore assessing the best therapy on a case-by-case basis.

In conclusion, although the use of DOACs appears from preliminary studies to be effective and safe in the treatment of CVT, additional studies are needed. This may prove difficult, however, since CVT remains a rare disease with a very high clinical and aetiological variability.

## Figures and Tables

**Figure 1 jcm-13-04730-f001:**
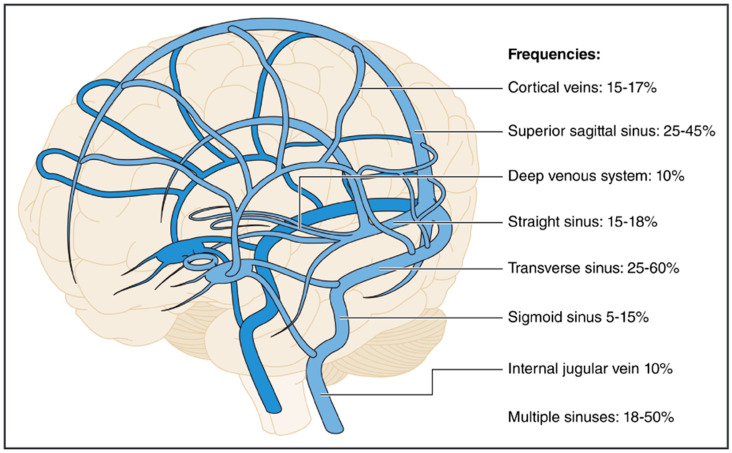
Anatomy of the cerebral venous system and distribution of CVT [4].

**Figure 2 jcm-13-04730-f002:**
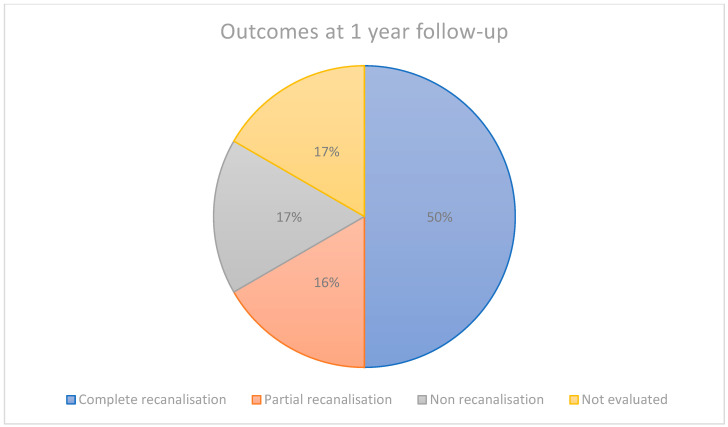
Percentage of complete, partial and non-recanalisations in our study patients.

**Table 1 jcm-13-04730-t001:** Predisposing factors of CVT.

Predisposing Factors	Transient	Chronic
Gender-specific conditions	Pregnancy and puerperium, oestrogen-containing contraceptives	
Other morbidity	Head and neck infections, COVID-19, sepsis, head trauma	Obesity, malignancies
Autoimmune and inflammatory diseases		Antiphospholipid antibody syndrome, systemic lupus erythematosus, Behçet’s disease, chronic inflammatory bowel disease
Haematological disorders		Mutated JAK V617F myeloproliferative diseases, paroxysmal nocturnal haemoglobinuria and haemoglobinopathies
Medications	Corticosteroids, L-aspariginase, thalidomide-analogues, tamoxifene	
Hereditary thrombophilia		Factor V Leiden mutation, prothrombin polymorphism (G20210A mutation) and congenital antithrombin, coagulative protein C and protein S deficiency

**Table 2 jcm-13-04730-t002:** Descriptive table of our patients’ characteristics.

ID	Gender	Age at Diagnosis (yo)	Aetiology	Other Risk Factors	Acute Treatment	Chronic Treatment	Outcome
1	F	25	JAK2+ MPN	Budd–Chiari syndrome, previous liver transplant	LMWH → AVK	AVK sine die	Partial recanalisation at 1 year
2	F	27	Oestroprogestinic therapy	Obesity	LMWH → AVK	LMWH prophylaxis in high thrombotic risk situations	Complete recanalisation at 1 year
3	F	27	Oestroprogestinic therapy	None	LMWH → AVK	LMWH prophylaxis in high thrombotic risk situations	Complete recanalisation at 6 months
4	F	31	Oestroprogestinic therapy	Isolated JAK2+	LMWH → AVK	DOAC prophylaxis (Rivaroxaban 10 mg die)	Complete recanalisation at 1 year
5	F	46	Idiopathic	Uterine fibromatosis; inflammatory thyroid disease	LMWH → AVK	AVK sine die	Non-recanalisation at 1 year; progression after COVID infection + INR not in range
6	F	47	Oestroprogestinic therapy	Obesity, arterial hypertension	LMWH → AVK	DOAC prophylaxis (Rivaroxaban 10 mg die)	Non-recanalisation at 1 year
7	M	48	Head trauma	None	LMWH → AVK	DOAC prophylaxis (Rivaroxaban 10 mg die)	Non-recanalisation at 6 months; complete recanalisation at 1 year
8	F	49	AT-III congenital deficiency	ENA +, congenital hypoplasia of the transverse sinus	LMWH → AVK	AVK sine die	Partial recanalisation at 1 year
9	F	57	Idiopathic	Obesity, arterial hypertension, previous melanoma	LMWH → AVK	DOAC prophylaxis (Apixaban 2.5 mg bid)	Complete recanalisation at 1 year
10	M	57	JAK2+ MPN	Arterial hypertension, previous ischemic stroke	LMWH → AVK	AVK sine die	Non-recanalisation at 6 months; not evalueted at 1 year yet
11	M	59	FV Leiden mutation	Crohn’s disease	LMWH → AVK	Not evaluated yet	Not evaluated yet
12	F	61	Idiopathic	None	LMWH	DOAC prophylaxis (Apixaban 2.5 mg bid)	Complete recanalisation at 6 months

MPN: myeloproliferative neoplasms; LMWH: low molecular weight heparin; AVK: anti-vitamin K; LMWH → AVK: LMWH imbricated with AVK; DOAC: direct oral anticoagulant; AT-III: anti-thrombin III; ENA: extractable nuclear antigen autoantibodies.

**Table 3 jcm-13-04730-t003:** Clinical manifestations and sinus involvement. MRA: Magnetic Resonance Angiography; CT: computed tomography.

ID	Symptoms	Diagnostic Imaging	Involved Sinus
1	Cervicalgia with progressive functional impotence	MRA	Superior sagittal sinus, torcular Herophili, sigmoid sinus, transverse sinus, jugular gulf
2	Severe headache not responsive to NSAIDs, photophobia, emesis	CT angiography	Right sagittal and transverse sinuses, right jugular vein
3	Persistent headache, dizziness, muffled hearing and tinnitus	CT angiography	Left transverse and straight sinuses, left vein of Galen, left jugular vein
4	Persistent headache, mild aphasia, drowsiness	CT angiography and MRA	Left transverse and sigmoid sinuses, left vein of Labbè, left jugular vein + SAH
5	Persistent right migraine, nausea, photophobia	MRA	Right transverse sinus
6	Vertiginous syndrome, otalgia, thunderclap left headache	MRA	Left transverse and sigmoid sinuses, left internal jugular, right transverse and straight sinuses
7	Left hearing loss and vertiginous syndrome	MRA	Left sigmoid sinus, left jugular gulf
8	Asthenia, widespread cramp-like pain in the lower limbs and decreased visus	CT angiography and MRA	Left sigmoid sinus and left internal jugular vein
9	Intense headache, fleeting hypoaesthesia and paresthesias of the right upper limb and right foot, buccal rima deviation	MRA	Superior sagittal sinus
10	Asthenia, disequilibrium, decreased visus	MRA	Right jugular gulf, right sigmoid and transverse sinuses
11	Headache, speech and motor impairment	CT angiography and MRA	Superior sagittal sinus, right transverse and sigmoid sinuses
12	Headache, nausea, transient sensory defect	CT angiography	Superior sagittal sinus, bilateral transverse sinuses

## Data Availability

No new data were created or analyzed in this study. Data sharing is not applicable to this article.

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
