# Peer review of "Cerebral Vein Thrombosis and Direct Oral Anticoagulants: A Review"

_jcm, 2024, doi:10.3390/jcm13164730_

Round 1

Reviewer 1 Report

Comments and Suggestions for Authors

- It is necessary to list the most common causes and expand the range of clinical symptoms and signs of the disease!

- Is DOAC the therapy of choice in all conditions and what in the population of patients with chronic kidney disease or treated hemodialysis!

- What is the advantage of DOAC compared to NOAC?

- What to do in centers that do not have the ability to determine anti Xa?

Author Response

Comments 1: It is necessary to list the most common causes and expand the range of clinical symptoms and signs of the disease!

Response 1: We agree with this comment. Therefore, I added an image of the anatomy of the cerebral venous system and distribution of CVT (page n 1, paragraph 1.2, lines 41, 42 and figure in page 2, paragraph 1.2 and line 50). I have also added predisposing factors for cerebral venous thrombosis and added a table summarising them (pages 2 and 3, paragraph 1.2, lines 68, 69, 73, 74, 77, 78, 88). Finally, I have implemented the paragraph on symptoms and clinical manifestations of CVT (page 3 and 4, paragraph 1.3, lines 91 and 92 and 97 – 109).

Comments 2: Is DOAC the therapy of choice in all conditions and what in the population of patients with chronic kidney disease or treated hemodialysis!

Response 2: This is a very important aspect to consider, and I therefore thank you for pointing it out. DOACs in recent years had a big expansion in their use also in patient populations underrepresented in registration trials. For this reason, studies on the use of DOACs in particularly high-risk populations have increased exponentially over the last period, considering the various factors that could affect the efficacy and safety of DOAC-based therapy. For patients with chronic kidney disease, recommendations are for dose adjustment based on creatinine clearence, taking into account the different renal clearance of each of the drugs. Indeed, apixaban is the drug with the lowest renal clearence (27%), followed by rivaroxaban (35%), edoxaban (50%) and dabigatran (80%). According to FDA prescriptive guidance, apixaban does not require dose adjustment based on renal impairment alone, but at least two of the following conditions must be present: age ≥ 80 years, body weight ≤ 60 kg, serum creatinine ≥ 1.5 mg/dl. In Europe, however, the recommendations are for a standard dose if creatinine’s clearence (CrCl) > 30 ml/min, dose adjustment if CrCl between 15 and 30 ml/min, and avoidance if CrCl < 15 ml/min. Rivaroxaban should be reduced if CrCl is between 15 and 50 ml/min and avoided if CrCl < 15 ml/min. Edoxaban requires dose adjustment in case of body weight < 60 kg, CrCl 15 - 49 ml/min or concomitant use of a potent p-glycoprotein inhibitor. The recommendations for dabigatran are to use a reduced dose in patients at increased risk of bleeding, and in the USA 75 mg bid is approved if CrCl 15 - 29 ml/min. Finally, with regard to the use of DOACs in haemodialysis, studies are limited and most of them focus on the use of apixaban, which would appear to be safe and effective in this particular class of patients. I have added these important aspects in the manuscript and references to them are ref. 41, 42 and 43.

Comments 3: What is the advantage of DOAC compared to NOAC?

Response 3: Thank you for highlighting this concept. DOAC and NOAC are acronyms used to refer to the same class of drugs, namely direct oral anticoagulants or non-vitamin K dependent oral anticoagulants. The acronym NOAC is also used to refer to the new oral anticoagulants, which is why this acronym is now used to a lesser extent, considering the well-established use of these drugs. The drugs indicated by the acronyms DOAC or NOAC currently in use are: apixaban, rivaroxaban, edoxaban and dabigatran.

Comments 4: What to do in centers that do not have the ability to determine anti Xa?

Response 4: Thank you for pointing this out. Actually, as DOACs have predictable pharmacokinetics and pharmacodynamics at fixed doses, their monitoring is not indicated. Nevertheless, in some emergency situations and/or special patient populations, measuring the concentration or activity of DOACs may be useful. There are some qualitative and quantitative tests that are, however, only available in specialised laboratories and, moreover, do not have an international standard for calibration that would allow them to be standardised for monitoring purposes. These tests therefore have no established clinical role and are rather used in clinical trials and for research purposes. Screening coagulation assays such as prothrombin time (PT), activated partial thromboplastin time (APTT), and thrombin time (TT) are routinely available in all laboratories and can therefore be useful in emergency situations, always bearing in mind the unreliability of these parameters during DOAC therapy. Indeed, the variability of results can be important depending on the sensitivity of the reagent to the DOAC and, for this reason, these tests can be used as a first-line assessment but not for monitoring the anticoagulant activity of DOACs. Under special conditions, quantitative tests such as the anti-factor Xa activity assay can be used to deduce a possible excess or deficiency of anticoagulant activity but as they are not approved for DOAC monitoring, they must always be related to clinical aspects. It is therefore vital to use other parameters for monitoring, including signs and symptoms of bleeding, platelet count, kidney and liver function.

Reviewer 2 Report

Comments and Suggestions for Authors

Venous thrombosis is a serious problem in modern medicine. Research in this area is very important and necessary since the prevalence of this disease is high and has a wide age range. In this regard, the topic of the manuscript proposed by the authors is undoubtedly relevant.

However, authors should structure their manuscript more clearly.

For example: subsections of the manuscript - 2.1, 2.2, 2.3 and 2.4 are too small and do not contain significant information, the authors should combine them into one large section.

Also, for greater clarity of the results obtained by the authors, it is recommended to add a diagram to the manuscript reflecting the percentage of complete and partial recanalization processes in the observed patients.

The question also arises:

Was there any concomitant chronic pathology in the observed group of patients?

Did they take any pills for these illnesses? All this can also influence the course of the disease being studied.

Since the manuscript does not contain a clinical description of the study group, the authors are recommended to add a section describing the clinical manifestations of the established diagnosis (patient complaints, physical examination data, cardiovascular and nervous system...).

 It is also recommended that authors, for clarity, add the results of MRI diagnostics or other studies that used to establish the diagnosis.

All this will make the manuscript more meaningful and understandable for clinicians.

Author Response

Comment 1: Subsections of the manuscript - 2.1, 2.2, 2.3 and 2.4 are too small and do not contain significant information, the authors should combine them into one large section.

Response 1: We agree with this comment. Therefore, I have implemented each section with some information regarding the mechanism of action, metabolism, dosage and possible dosage adjustments and peculiarities of the various DOACs. I hope the information will be more complete this way (page 6, paragraph 2.1, lines 228-236; page 6, paragraph 2.2, lines 250-261; page 7, paragraph 2.3, lines 271-278; page 7, paragraph 2.4, lines 293-298).

Comment 2: it is recommended to add a diagram to the manuscript reflecting the percentage of complete and partial recanalization processes in the observed patients.

Response 2: Agree. I have accordingly added a diagram at page 11.

Comment 3: Was there any concomitant chronic pathology in the observed group of patients?

Response 3: Thank you for pointing this out. Actually Table 1 already described the additional risk factors for the onset of CVT presented by the patients, but in any case, thanks to your observation, I felt it was clearer to implement in the text the description of the most relevant comorbidities of the patients studied (page 9, paragraph 3, lines 321-327).

Comment 4: Did they take any pills for these illnesses? All this can also influence the course of the disease being studied.

Response 4: Agree. I have accordingly described the chronic pre-existing therapies of patients in page 9, paragraph 3, lines 327-332.

Comment 5: Since the manuscript does not contain a clinical description of the study group, the authors are recommended to add a section describing the clinical manifestations of the established diagnosis (patient complaints, physical examination data, cardiovascular and nervous system...). It is also recommended that authors, for clarity, add the results of MRI diagnostics or other studies that used to establish the diagnosis

Response 5: Thank you for your comment, I found this to add useful information to the manuscript. Therefore, I added a brief description of the patients' onset symptoms, the diagnostic test used and the location of the cerebral thrombosis (page 9, paragraph 3, lines 333-338). For the sake of clarity, I added a descriptive table in this regard (table 3).

Round 2

Reviewer 1 Report

Comments and Suggestions for Authors

No